# Recurrent Esthesioneuroblastoma: Long-Term Outcomes of Salvage Therapy

**DOI:** 10.3390/cancers15051506

**Published:** 2023-02-28

**Authors:** Garrett Ni, Carlos D. Pinheiro-Neto, Ehiremen Iyoha, Jamie J. Van Gompel, Michael J. Link, Maria Peris-Celda, Eric J. Moore, Janalee K. Stokken, Mauricio Gamez, Garret Choby

**Affiliations:** 1Department of Otolaryngology-Head and Neck Surgery, Lewis Katz School of Medicine, Temple University, Philadelphia, PA 19122, USA; 2Department of Otolaryngology-Head and Neck Surgery, Mayo Clinic School of Medicine, Rochester, MN 55905, USA; 3Department of Neurologic Surgery, Mayo Clinic, Rochester, MN 55905, USA

**Keywords:** esthesioneuroblastoma, olfactory neuroblastoma, overall survival, progression-free survival, recurrence, risk factors, long-term outcome, sinonasal malignancy, metastasis

## Abstract

**Simple Summary:**

Esthesioneuroblastoma (ENB) recurrence is common; however, data regarding patient outcomes after recurrence is still lacking. This study examines patient outcomes after ENB recurrence using 143 patients with ENB and describes the patterns of salvage therapeutic interventions. Our study found the 5-year overall survival after ENB recurrence to be 63% and the 5-year progression-free survival to be 56%. The mean time to develop a secondary recurrence was significantly shorter than the mean time to develop the first recurrence. Additionally, patients who developed secondary recurrences were, on average, significantly older at the time of their primary tumor diagnosis.

**Abstract:**

Introduction: Esthesioneuroblastoma (ENB) is a rare malignant neoplasm arising from the olfactory epithelium of the cribriform plate. Although survival is excellent with a reported 5-year overall survival (OS) of 82%, recurrence is frequent and occurs in 40–50% of cases. This study investigates the characteristics of ENB recurrence and the subsequent prognosis of patients with recurrence. Methods: The clinical records of all patients diagnosed as having ENB with subsequent recurrence at a tertiary hospital from 1 January 1960 to 1 January 2020 were retrospectively reviewed. Overall survival (OS) and progression-free survival (PFS) were reported. Results: A total of 64 out of 143 ENB patients had recurrences. In total, 45 out of 64 recurrences met the inclusion criteria and were included in this study. From these, 10 (22%) had a sinonasal recurrence, 14 (31%) had an intracranial recurrence, 15 (33%) had a regional recurrence, and 6 (13%) had a distal recurrence. The average interval from initial treatment to recurrence was 4.74 years. There were no differences in rates of recurrence with respect to age, sex, or types of surgery (endoscopic, transcranial, lateral rhinotomy, and combined). The time to recurrence was shorter for Hyams grades 3 and 4 compared to Hyams grades 1 and 2 (3.75 years vs. 5.70 years, *p* < 0.05). Patients with recurrence limited to the sinonasal region had a lower overall primary Kadish stage compared to recurrences beyond the sinonasal region (2.60 vs. 3.03, *p* < 0.05). A total of 9 (20%) out of 45 patients developed secondary recurrence. Following recurrence, the subsequent 5-year OS and PFS were 63 and 56%, respectively. The mean time to secondary recurrence after treatment of the primary recurrence was 32 months, which was significantly shorter than the time to primary recurrence (32 months vs. 57 months, *p* = 0.048). The mean age of the secondary recurrence group is significantly older than the primary recurrence group (59.78 years vs. 50.31 years, *p* = 0.02). No statistically significant differences were observed between the secondary recurrence group and the recurrence group in terms of their overall Kadish stages or Hyams grades. Conclusions: Following an ENB recurrence, salvage therapy appears to be an effective therapeutic option with a subsequent 5-year OS of 63%. However, subsequent recurrences are not infrequent and may require additional therapy.

## 1. Introduction

Esthesioneuroblastoma (ENB) recurrence has been estimated to be around 30 to 60% after successful treatment of the primary tumor [1,2,3,4,5]. Recurrent disease is usually locoregional and tends to have a long interval to relapse with a mean of 6 years [6].

While there are no universally accepted treatment guidelines for ENB due to the rarity of the disease and the lack of clinical trials, it is common to involve dual-modality with both surgery and radiotherapy for early-stage diseases and the addition of chemotherapy for advanced disease or worrisome prognostic features, such as a high Hyams grade or positive surgical margins. In select cases of locally advanced disease in which surgical resection with negative margins is unlikely, induction chemotherapy may be an option as part of an organ-preservation surgical strategy or to reduce tumor burden to help to achieve negative margins [7,8,9,10].

The most widely used staging system for ENB is the Kadish system which classifies the primary tumor based on the extent of local tumor anatomic involvement. The Kadish system has been modified (mKadish) by Morita et al., and ranges from A to D with stage A involving the nasal cavity alone, stage B extending to the paranasal sinuses, stage C involving the cribriform plate, the skull base, the orbit, or intracranial cavity, and stage D involving the neck or any distal metastasis [11]. Additionally, ENBs are also graded based on pathological features using the Hyams grading system [12]. Hyams grades range on a scale of one to four based on the histologic findings of the tumor biopsy and confer the aggressiveness of the tumor [13]. Hyams grading is commonly categorized into low-grade Hyams (LGH; Hyams I–II) and high-grade Hyams (HGH; Hyams III–IV) when characterizing tumor behavior and prognoses [14].

Prior studies demonstrated that ENBs with advanced Kadish stages have an increased risk of distant failure and treatment complications attributed to more extensive surgical resection and aggressive adjuvant therapies [15,16]. Similarly ENBs with HGH generally have lower overall survival rates due to an increased likelihood of metastasis to the neck or distant anatomic sites [12]. 

Although the overall survival (OS) of ENB patients is generally favorable compared to other sinonasal malignancies, recurrence is quite common and tends to occur in a delayed fashion, often 5 years or more after initial treatment. When recurrence occurs, the treatment guidelines are not well established but mainly consist of salvage surgery, targeted radiotherapy, and/or chemotherapy. However, little has been published on the long-term subsequent outcomes of ENB patients treated with salvage surgery for recurrence. Thus, the purpose of this study is to identify recurrence patterns of ENB, describe salvage therapeutic interventions, characterize subsequent long-term survival, and analyze the factors associated with survival and secondary recurrences. 

## 2. Materials and Methods

This retrospective cohort study was approved by the Mayo Clinic Institutional Review Board #18-001238. The clinical records of all patients diagnosed with ENB at the Mayo Clinic from August 1960 to December 2019 were retrospectively reviewed. Exclusion criteria included patients missing clinical or treatment records, documentation of Kadish stages, Hyams grades, or recurrence data. 

All patients included in the study were cared for by a multidisciplinary team consisting of otolaryngologists—head and neck surgeons, radiation oncologists, medical oncologists, neurosurgeons, neuroradiologists, and neuro-ophthalmologists, as needed. Computed tomography and, later in the series, magnetic resonance imaging were used to assess the full extent of each tumor before the commencement of treatment. The patients’ tumors were staged based on clinical and radiologic presentations according to the Kadish stage. 

Subgroup analysis was conducted using Cox proportional hazards model, Student’s *t*-test, and Fisher exact test. The Kaplan–Meier method was used to estimate the probability of recurrence or death. Death caused by the disease was treated as an endpoint for disease-specific survival, while other deaths, such as postoperative deaths, were treated as censored observations. The subsequent period was defined as the time from recurrence to the time of last follow-up or death. 

All-cause death was defined as any patient with a documented date of death, regardless of the reason. Cancer-specific death was defined as any patient with a documented date of death that also had a cause of death listed as “Dead from disease” or “Dead of disease”. Cancer progression was defined as the earliest of 2 possible dates, secondary recurrence, or cancer-specific death. If a patient did not meet the requirements for an event in each outcome, they were censored at the time of the last follow-up. Associations of demographic and clinical features with each outcome were evaluated using univariable Cox proportional hazards regression models and summarized using hazard ratios (HRs) and 95% confidence intervals (CIs). Statistical analyses were performed using SAS version 9.4 (SAS Institute; Cary, NC, USA), and plots were generated using R version 3.6.2 (R Foundation for Statistical Computing; Vienna, Austria). All tests were two-sided, and *p*-values < 0.05 were considered statistically significant.

## 3. Results

### 3.1. Patient Characteristics

Of the 143 included ENB patients, 64 developed recurrent disease. A total of 45 out of 64 recurrences met the inclusion criteria and were included in this study. There were 17 (38%) males and 28 (62%) females. The most common surgery performed for primary ENB was the transcranial approach (62%). This is followed by the combined transcranial and endonasal approach (18%), purely endoscopic approach (13%), and lateral rhinotomy (5%). One (2%) patient did not undergo surgery because the tumor was deemed unresectable. In total, 41 (91%) out of 45 patients underwent radiation therapy as part of their primary ENB treatment. The distribution of mKadish stages in our cohort was 2.2% stage A, 13.3% stage B, 80.0% stage C, and 4.4% stage D. The distribution of Hyams grades in our cohort was 8.8% grade 1, 37.7% grade 2, 40.0% grade 3, and 13.3% grade 4 (Table 1). 

The average interval from the initial treatment of the primary tumor to recurrence was 4.8 years (interquartile range = 4.3 years). The mean age of patients at the time of ENB recurrence in our sample was 56.8 years (range 30.8 to 85.1 years) (Table 1). The median follow-up time for these patients was 76.6 months (interquartile range of 25.5–153 months).

In terms of location of recurrence, 12 (26.7%) patients developed a local recurrence with a median time to recurrence of 57.6 months. Fourteen (31.1%) patients developed an intracranial recurrence with a median time to recurrence of 39.6 months. Fifteen (33.3%) patients developed cervical lymph node metastasis, with a median time to metastasis of 70 months. Six (13.3%) patients developed distant metastasis, including bone, liver, orbit, and oral cavity, with a median time to metastasis of 47.2 months (Table 2). 

The mean time to recurrence was shorter for HGH compared to LGH (3.75 years vs. 5.70 years, *p* = 0.02) (Table 3). Patients with recurrence limited to the sinonasal region had a lower overall primary Kadish stage compared to recurrences beyond the sinonasal region (2.50 vs. 2.95, *p* = 0.01) (Table 4).

Cervical lymph node metastasis showed a significantly lower Hyams grade compared to distant metastasis (2.4 vs. 3.14, *p* = 0.01). No significant differences in Hyams grades or Kadish stages were observed between all other recurrence locations. No differences were found with regard to age, sex, time to recurrence, and margin status across all recurrence locations.

### 3.2. Salvage Therapy

A total of 39 out of 45 patients had adequate documentation of their salvage therapy. Seventeen (44%) patients received single-modality therapy, followed by dual-modality therapy in twelve (30%), and three or more modalities therapy in ten (26%). Overall, 27 (69%) patients received surgical treatment as part of their salvage therapy. The most common surgical therapy received was neck dissection with 20 (51%) patients. Among other treatment modalities, 22 (56%) patients received radiation therapy, 8 (20%) received gamma knife surgery, and 10 (26%) received chemotherapy (Table 5). In terms of chemotherapy agents, 70% received platinum-based chemo, 40% received taxanes, 50% received a topoisomerase inhibitor, and 30% received an alkylating agent. 

### 3.3. Secondary Recurrence

Among the 45 patients with ENB recurrence, 9 (20%) patients developed a subsequent secondary recurrence. The mean time to secondary recurrence after treatment of the primary recurrence was 32 months, which is significantly shorter than the time to primary recurrence (32 months vs. 57 months, *p* = 0.048). In terms of the location of the secondary recurrence, the distribution was four (44%) sinonasal, one (11%) intracranial, three (33%) regional, and two (22%) distal. Three (33%) patients had secondary recurrences that shared the same location as the primary recurrence. The locations of secondary recurrences are shown in Table 6. The secondary recurrence group were predominantly male (8, 88%), although no statistical significance was found compared to the group without secondary recurrence. The mean age of the secondary recurrence group is significantly older than the primary recurrence group (59.78 years vs. 50.31 years, *p* = 0.02). No statistically significant differences were observed between the secondary recurrence group and the recurrence group in terms of their overall Kadish stages or Hyams grades. There were no statistically significant differences in terms of the secondary recurrence rate between patients treated prior to 2000 and those treated after 2000. 

### 3.4. Survival Analysis

Following treatment for the primary tumor, the subsequent 5-year OS was 86%, and the 5-year PFS was 41%. The 1-, 2-, 5-, and 10-year cumulative incidence for each outcome is shown in Table 7. 

Figure 1 show this graphically.No significant differences were found between patients with high and low Hyams grades in any of the outcomes. Figure 2 shows all cause death and cancer progression following recurrence.

With respect to survival after esthesioneuroblastoma recurrence and salvage therapy, the subsequent 5-year OS was 63%, and the 5-year PFS was 56%. The 1-, 2-, 5-, and 10-year cumulative incidence for each outcome is shown in Table 8.

Table 9 reports the tests for the association between each outcome and demographic and clinical factors as summarized using hazard ratios and confidence intervals. No significant associations were found.

## 4. Discussion

While there are a handful of studies looking at the long-term outcomes of primary ENB, less is known about the outcomes of ENB recurrences. In this cohort, the regional recurrences were associated with a lower Hyams grade compared to distal metastases. The mean time to recurrence was shorter for HGH compared to LGH, and patients with recurrence limited to the sinonasal region had a lower overall primary Kadish stage compared to recurrences beyond the sinonasal region. A handful of recurrences went on to develop a subsequent secondary recurrence, which was associated with increased age. Patients with ENB recurrences appear to carry a relatively favorable prognosis with a subsequent 5-year OS of 63% following salvage therapy. 

With regard to the location of initial recurrence, our study demonstrated a very similar recurrence pattern to that reported by prior studies. The regional recurrence in our sample was 33.3%, which is also consistent with reports from previous studies [17,18,19,20,21]. The high percentage of regional recurrence is likely attributed to the high prevalence of advanced Kadish stages in our sample, which is known to be a risk factor for neck lymph node metastasis. Furthermore, at the Mayo Clinic, patients with N0 necks do not receive elective neck irradiation (ENI), which may have contributed to the increase rates of regional metastasis. There is currently no consensus regarding ENI for N0 necks in ENB patients. Prior studies showed ENI improves 5-year PFS but does not appear to offer OS benefits [22,23,24]. A recent multicenter study also showed that ENI may reduce rates of delayed nodal recurrence [25]. However, without compelling evidence of clear OS benefits, it is important to take in consideration the side effects of radiation and its potential impact on patients’ quality of life. The side effects of ENI, including dysphagia, xerostomia, mobility restriction, hypothyroidisms, carotid stenosis, and carotid blowout syndrome are well reported in the literature [26,27]. Additionally, salvage neck dissection has been shown to be a viable therapy in patients with delayed regional metastases with excellent subsequent survival and regional disease control [28]. 

ENB recurrence time varies widely with a reported range between 9 and 251 months [29]. Our study found that the time to recurrence was shorter for HGH compared to LGH (3.75 years vs. 5.70 years, *p* < 0.05). Although we did not find a significant difference in recurrence time by the location of the recurrence, studies have demonstrated a correlation between HGH with distant recurrences, which tend to occur sooner than local recurrences [6,30]. These findings are likely due to the more aggressive cellular behavior of HGH and hence earlier and more distant recurrence. Additionally, we found that cervical neck metastases were associated with lower Hyams grade compared to distant metastases.

Following a recurrence, the subsequent 5-year OS was 63%, and the PFS was 56%. Recurrent ENB shares a similar 5-year OS and PFS to that of primary ENB reported by prior studies [11,31]. The trend of the OS at years 1, 2, 5, and 10 between HGH and LGH was consistently higher for LGH; however, the comparison was statistically insignificant. Although there were no studies examining survival with respect to Hyams grade in ENB recurrence, prior studies examining survival in primary ENB suggested that a higher Hyams grade generally correlates with worse OS and PFS [32,33,34,35].

Secondary recurrence has a shorter time to recurrence compared to primary recurrence in our cohort. The reason for this is unclear but could possibly be explained by a couple of reasons. Although primary ENB therapy tends to be multimodal, salvage treatment tends to be unimodal, and re-irradiation is sometimes not an option. Therefore, it is possible that treatment options were limited for the salvage therapy and that some patients were not re-irradiated for their recurrence, which can explain the faster recurrence. Prior studies have shown the addition of radiotherapy resulted in significantly better OS and DFS compared to surgery alone and other treatment modes [36]. Patients in the secondary recurrence group were also significantly older and maybe predisposed to sooner recurrence due to waning immunity.

There are several limitations to this study. As a retrospective study that spanned over the course of many years, the treatment modalities, guidelines, and understanding of the disease have changed significantly during this period. This potentially introduces inconsistencies in our data as our cohort was not homogenously managed. Our ENB recurrence sample was mainly Kadish stage C (80%); whether this was a result of selection bias or chance, it inevitably skewed the characteristics of our sample, and our results may not be generalizable and did not allow for subgroup analysis between different Kadish stages. Lastly, although this study represents one of the largest retrospective cohort studies, it is still restricted by the number of patients with recurrence, which did not allow for a meaningful comparison between some subgroups. 

## 5. Conclusions

ENB recurrences are common and often intracranial and regional. Following ENB recurrence, salvage therapy appears to be an effective therapeutic option with a subsequent 5-year OS of 63%. However, subsequent recurrences are not infrequent and may require additional therapy.

## Figures and Tables

**Figure 1 cancers-15-01506-f001:**
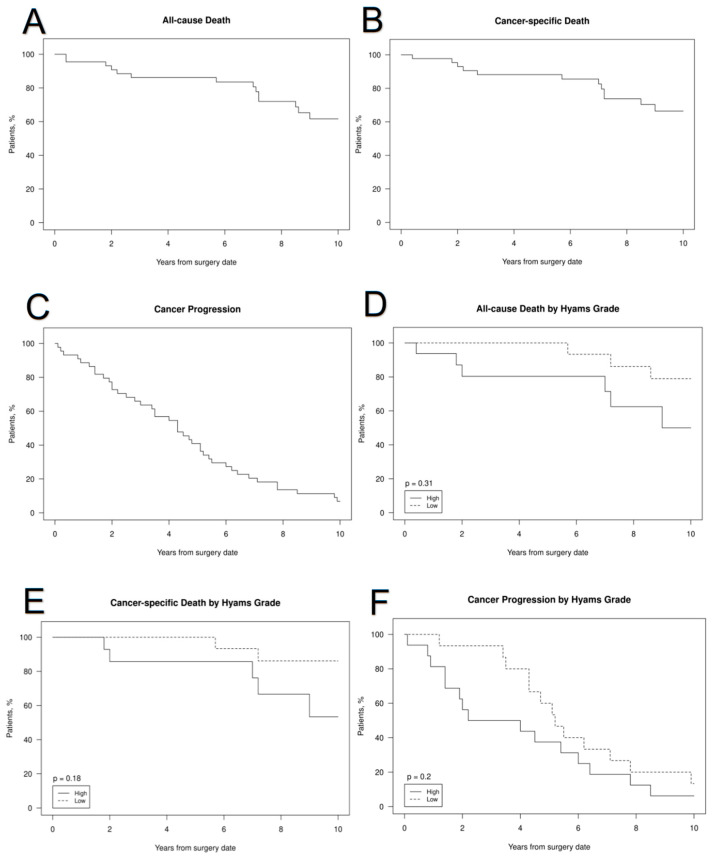
All-cause mortality from date of primary ENB diagnosis in patient with recurrences (**A**), cancer-specific mortality from date of surgery (**B**), cancer progression from date of surgery (**C**), all-cause mortality from date of surgery between HGH and LGH (**D**), cancer-specific mortality from date of surgery between HGH and LGH (**E**), cancer progression from date of surgery between HGH and LGH (**F**).

**Figure 2 cancers-15-01506-f002:**
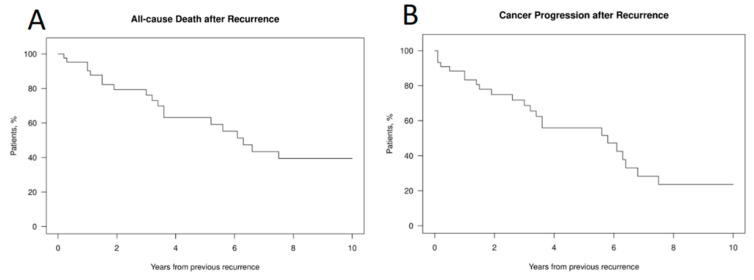
All-cause mortality after ENB recurrence (**A**) and cancer progression after recurrence (**B**).

**Table 1 cancers-15-01506-t001:** Patient demographics.

**Mean age at recurrence (years)**	56.83 (range: 30.8 to 85.1)
**Mean time to recurrence (years)**	4.74 (range: 0.08 to 15.91)
**Sex (%)**	
Male	17 (38)
Female	28 (62)
**Surgical approach for primary tumor (%)**	
Endoscopic	6 (13)
Transcranial	28(62)
Combined	8 (18)
Unresectable	1 (2)
Lateral rhinectomy	2 (4)
**Other treatments (%)**	
Radiation therapy	41 (91)
Gama knife surgery	9 (22)
Chemotherapy	27 (60)
**Hyams grade (%)**	
Grade 1	4 (9)
Grade 2	17 (38)
Grade 3	18 (40)
Grade 4	6 (13)
**Kadish stage (%)**	
Stage A	1 (2)
Stage B	6 (13)
Stage C	36 (80)
Stage D	2 (4)

**Table 2 cancers-15-01506-t002:** Recurrence by locations.

Location of Recurrence	Number of Patients (%)	Time to Recurrence (Month)
Sinonasal	10 (22.2%)	57.4
Intracranial ^a^	14 (31.1%)	39.6
Cervical lymph node(s)	15 (33.3%)	52.0
Distal organs ^b^	6 (13.3%)	57.2

^a^ Intracranial recurrences include dural metastases and brain; ^b^ distal organs included bone, liver, and oral cavity.

**Table 3 cancers-15-01506-t003:** Hyams grade and Kadish stage relation to recurrent time and local recurrence.

	Hyams Grade 1 and 2 (n = 21)	Hyams Grade 3 and 4 (n = 24)	*p*-Value
Time to recurrence (years)	5.70	3.75	<0.05

**Table 4 cancers-15-01506-t004:** Mean Kadish stage between sinonasal recurrence and non-sinonasal recurrence.

	Sinonasal Recurrence (n = 10)	Regional or Distal Recurrence (n = 35)	*p*-Value
Mean Kadish stage of primary tumors	2.60	3.03	<0.05

**Table 5 cancers-15-01506-t005:** Salvage therapy (n = 39).

**Modalities of therapy (%)**	
Single	17 (44)
Dual	12 (30)
Triple	10 (26)
**Surgical treatment (%)**	27 (69%)
**Surgical approach for recurrence (%)**	
Endoscopic	4 (10)
Transcranial	3 (8)
Other open surgery	7 (18)
Neck dissection	20 (51)
**Other treatments (%)**	
Radiation therapy	22 (56)
Gama knife surgery	8 (20)
Chemotherapy	10 (26)

**Table 6 cancers-15-01506-t006:** Esthesioneuroblastoma secondary recurrences.

Patient #	Surgery	Kadish	Hyams	Margin Status	Radiotherapy for Primary ENB	Primary Recurrence	Secondary Recurrence	Time to Secondary Recurrence (Months)	Vital Status
1	Transcranial	C	3	Unknown	Yes	L frontal lobe, dura, bone, and skin	R oral cavity, R submandibular gland, and L parotid	1	Deceased
2	Transcranial	C	2	Negative	Yes	R frontal dura and lobe	R maxillary sinus	82	Deceased
3	Transcranial	C	3	Negative	Yes	L frontal parasagittal dura	Bilateral neck and L parotid	32	Alive
4	Unresectable	C	2	Unknown	Yes	Sinonasal cavity	Sinonasal cavity	70	Deceased
5	Transcranial	C	2	Negative	No	R nasal cavity and R maxillary sinus	R neck	77	Alive
6	Transcranial	C	3	Positive	Yes	R neck	R parietal dura	4	Alive
7	Combined	C	3	Negative	Yes	L neck	L ethmoid sinus	5	Alive
8	Endoscopic	B	3	Positive	Yes	R neck	Retropharyngeal node	6	Alive
9	Transcranial	B	2	Unknown	Yes	Maxilla/oral cavity	L maxillary sinus and L neck	17	Deceased

**Table 7 cancers-15-01506-t007:** The 1-, 2-, 5-, and 10-year cumulative incidence of all-cause death, cancer-specific death, and cancer progression after primary tumor diagnosis.

	**All-Cause Death**		
Year	Overall	Hyams Grade	
		High	Low
1	5% (0–11%)	6% (0–17%)	0% (0–0%)
2	9% (0–17%)	20% (0–37%)	0% (0–0%)
5	14% (3–24%)	20% (0–37%)	0% (0–0%)
10	38% (20–53%)	50% (8–73%)	21% (0–40%)
	**Cancer–Specific Death**		
		High	Low
1	2% (0–7%)	0% (0–0%)	0% (0–0%)
2	7% (0–14%)	14% (0–31%)	0% (0–0%)
5	12% (2–21%)	14% (0–31%)	0% (0–0%)
10	34% (16–48%)	47% (3–71%)	14% (0–30%)
	**Cancer Progression**		
		High	Low
1	11% (2–20%)	19% (0–36%)	0% (0–0%)
2	27% (13–39%)	44% (13–64%)	7% (0–19%)
5	59% (42–71%)	63% (29–80%)	40% (9–60%)
10	93% (80–98%)	94% (58–99%)	87% (52–96%)

**Table 8 cancers-15-01506-t008:** The 1-, 2-, 5-, and 10-year cumulative incidence of all-cause death and cancer progression after salvage treatment of recurrence.

Year	All-Cause Death	Cancer Progression
1	10% (0–18%)	17% (5–27%)
2	21% (7–33%)	25% (10–38%)
5	37% (18–51%)	44% (25–59%)
10	61% (38–75%)	76% (51–89%)

**Table 9 cancers-15-01506-t009:** Univariable hazard ratios of demographic and medical features.

Parameter	All-Cause Death		Cancer-Specific Death		Cancer Progression	
	HR (95% CI)	*p*-Value	HR (95% CI)	*p*-Value	HR (95% CI)	*p*-Value
**Age**	0.97 (0.93, 1.01)	0.17	0.96 (0.92, 1.01)	0.10	0.98 (0.95, 1.00)	0.07
**Sex**						
F	Ref		Ref		Ref	
M	0.64 (0.23, 1.73)	0.38	0.55 (0.18, 1.67)	0.29	0.81 (0.35, 1.86)	0.62
**Surgery**		0.82		0.91		0.59
Endoscopic	5.49 (0.30, 100.93)	0.25	-		2.07 (0.34, 12.54)	0.43
Transcranial	2.51 (0.31, 20.21)	0.39	-		1.54 (0.35, 6.77)	0.57
Combined	1.95 (0.18, 21.61)	0.59	-		2.90 (0.55, 15.22)	0.21
Unresectable	1.48 (0.09, 24.39)	0.78	-		4.30 (0.36, 51.42)	0.25
Lateral rhinotomy	Ref		Ref		Ref	
**Resection**		1.00		0.89		0.69
1	Ref		Ref		Ref	
2	-		-		0.51 (0.07, 3.86)	0.52
3	1.02 (0.23, 4.63)	0.98	0.60 (0.08, 4.78)	0.63	1.38 (0.41, 4.63)	0.61
**GKS gamma knife**						
0	Ref		Ref		Ref	
1	0.48 (0.11, 2.11)	0.33	0.68 (0.15, 3.04)	0.61	0.63 (0.24, 1.68)	0.36
**Final margins**		0.47		0.85		0.82
Negative	Ref		Ref		Ref	
Positive	0.74 (0.09, 6.09)	0.78	0.77 (0.09, 6.38)	0.81	0.89 (0.29, 2.72)	0.84
Unknown	1.70 (0.65, 4.45)	0.28	1.28 (0.43, 3.85)	0.66	1.25 (0.55, 2.83)	0.60
**Hyams grade**						
Low	Ref		Ref		Ref	
High	1.63 (0.63, 4.24)	0.31	2.07 (0.71, 6.05)	0.19	1.60 (0.78, 3.29)	0.20

## Data Availability

Data used in this article are stored on the Mayo Clinic Esthesioneuroblastoma database.

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
