# Peer review of "Recurrent Esthesioneuroblastoma: Long-Term Outcomes of Salvage Therapy"

_cancers, 2023, doi:10.3390/cancers15051506_

Round 1

Reviewer 1 Report

I thank the authors for their efforts and interesting results which will be a valuable addition to the literature. A few minor suggestions:

-In the results please consider describing what made the 2 unresectable cases unresectable, and in the discussion consider providing what your expert group thinks are the limits of offering definitive treatment. 

-Since the included cohort was treated in the span o almost 60 years during which there may have been important technological or imaging/therapeutic changes, consider comparing recent vs non-recent patients (e.g. let's say treated since 2000 vs before, something reasonable) and comparing their treatment method and survival, and if different, discuss in the discussion why there may have been changes, and what can similarly contribute to better treatments/survivals in the future. 

Author Response

Reviewer #1

Thank the authors for their efforts and interesting results, which will be a valuable addition to the literature. A few minor suggestions: In the results, please consider describing what made the 2 unresectable cases unresectable, and in the discussion consider providing what your expert group thinks are the limits of offering definitive treatment. 

The 1 unresectable case was deemed unresectable due to direct tumor involvement of the cavernous sinus, internal carotid artery and/or superior orbital fissure. Added to the result section.

Since the included cohort was treated in the span of almost 60 years during which there may have been important technological or imaging/therapeutic changes, consider comparing recent vs non-recent patients (e.g. let's say treated since 2000 vs before, something reasonable) and comparing their treatment method and survival, and if different, discuss in the discussion why there may have been changes, and what can similarly contribute to better treatments/survivals in the future. 

This is a very good point.  This study mainly examines outcomes after recurrence. When comparing the probability of secondary recurrences, we did not find any significant difference in those treated before 2000 and after 2000. However, due to the small number of patients with secondary recurrences, the comparison was likely insufficiently powered. This was added to the result section.

Reviewer 2 Report

This manuscript describes a retrospective cohort study of patients diagnosed with recurrent estheioneuroblastoma within the Mayo Clinic Health System from August 1960 to December 2016.  Forty five patients met the inclusion criteria and were included.  The quality of the analysis is adequate and the presentation is good.  Although the authors do partially address this, a retrospective review of patients over a 60 year period is fraught with inhomogenous variables such as quality of imaging, interval of follow up, evolving surgical techniques, use of surveillance imaging, radiation standards and quality metrics. This needs to be clearly articulated.  Two additional small queries: What percentage of recurrences were symptomatic versus asymptomatic at recurrence? What were the chemotherapy agents used in the 26% of patients at first recurrence?

Page 2, last paragraph: Fisher's exact test.

Author Response

Reviewer #2

This manuscript describes a retrospective cohort study of patients diagnosed with recurrent estheioneuroblastoma within the Mayo Clinic Health System from August 1960 to December 2016.  Forty five patients met the inclusion criteria and were included.  The quality of the analysis is adequate and the presentation is good.  Although the authors do partially address this, a retrospective review of patients over a 60 year period is fraught with in homogenous variables such as quality of imaging, interval of follow up, evolving surgical techniques, use of surveillance imaging, radiation standards and quality metrics. This needs to be clearly articulated.  Two additional small queries:

What percentage of recurrences were symptomatic versus asymptomatic at recurrence?

This is a good point and something that we had hoped to analyze.  Unfortunately, this was not documented in the Mayo Clinic database and a large portion of the patients were treated prior to the current EMR and was not documented.

What were the chemotherapy agents used in the 26% of patients at first recurrence?

Within the chemotherapy (10 patients) 70% received platinum-based chemo, 40% received taxanes, 50% received an topoisomerase inhibitor based chemo, and 30% received an alkylating agent. This was added to the result section under salvage therapy.

Reviewer 3 Report

This study is a retrospective study for recurrent Ethesioneuroblastoma.  There are a few concerns before publishing.

1. Please describe the reason of the difference of surgical procedure for recurrent disease.

2. The authors did not use Dulguerov classification. Why? Please describe the reason the authors used only Kadish classification.

Author Response

Reviewer #3

This study is a retrospective study for recurrent Ethesioneuroblastoma.  There are a few concerns before publishing.

  1. Please describe the reason for the difference of surgical procedure for recurrent disease.

This is a very relevant point.  Part of the explanation for different treatments for recurrent disease is because this is a long-term cohort study. Historically, transcranial approaches were most commonly utilized, occasionally paired with a lateral rhinotomy.  In the modern era, endoscopic craniofacial resection has become more standard in select cases, although combined open/endoscopic approaches remain important for disease involving dura over the orbit or planum.  Salvage therapies also varied depending on the extent of the recurrence. In contrast to localized recurrence in which single modality is generally employed, advanced recurrences generally require a much more robust regimen involving multiple treatment modalities including surgery, radiation, and chemotherapy. Within our cohort, 44% of the patients had localized recurrence and received single modality therapy.

  1. The authors did not use Dulguerov classification. Why? Please describe the reason the authors used only Kadish classification.

This is a good point.  Although the Kadish staging system is most commonly utilized, there are a total of four potential systems that can be utilized – Kadish, Morita-Kadish, Dulguerov and AJCC.  In our clinical practice, the Kadish system is most commonly used and we think this is reflective of most practices.  Although the Dulguerov system has value to delineate local tumors (i.e. T stage), this was not documented in most of our ENB records, especially for the cohort of patients in this study collected before the era of EMR.  Thus, we were somewhat limited to apply this staging system retroactively.